

# Catch fast and kill quickly: do tiger beetles use the same strategies when hunting different types of prey?

Tomasz Rewicz[*] and  Radomir Jaskuła[*]

Department of Invertebrate Zoology and Hydrobiology/Faculty of Biology and Environmental Protection, University of Lodz, Łódź, Poland
[*] These authors contributed equally to this work.

## ABSTRACT

**Background**. Tiger beetles (Coleoptera: Cicindelidae) are fast running predatory insects preying on different small insects and other terrestrial arthropods. Prey is located by sight and captured after short and fast pursuit interspersed with pause-and-look behaviour. At least some tiger beetle species can recognise the size and location of prey using memory, which probably allows them to achieve greater hunting success.

**Material and Methods**. Two eurytopic tiger beetle species known to occur in different types of habitat were used in the study: *Cicindela hybrida hybrida*, a very common central European beetle found even in artificial habitats such as sandy roads or gravel pits, and *Calomera littoralis nemoralis*, a species widely distributed in southern European countries and occurring on sandy sea beaches, in salt marshes, as well as on sandy banks of rivers and lakes. Both species are very similar in body size. Specimens used in the study were collected in the field and later tested in the laboratory. We checked whether tiger beetles use different hunting strategies when attacking prey of different sizes and abilities to escape as well as whether the sex of the studied species makes a difference in its hunting behaviour.

**Results**. The hunting strategies of both tiger beetle species consist of the following main phases: identification, pursuit (often with stops), attack, and optional release of the prey, and then the secondary attack, abandonment of the prey, or consumption of the prey. Considerable differences were noticed in hunting behaviour depending on the type of prey, its movement ability and escape potential. Caterpillars were attacked without pursuit, in the head or directly behind the head where a concentration of nerves and main muscles responsible for walking are located. Effective attacks on beetles were executed at the connection between the thorax and the abdomen. *Calomera littoralis* strongly preferred slow moving prey, while *Cicindela hybrida* preferred in equal measure slow moving prey and medium-sized fast moving prey. The experiment on the preferred size of prey indicated small beetles and small caterpillars as favoured by *Calomera littoralis*, while *Cicindela hybrida* preferred medium-sized fast moving prey and large caterpillars.

**Discussion**. The hunting behaviour of *Calomera littoralis* and *Cicindela hybrida* is complicated and includes a number of phases allowing to locate, capture and kill the prey. Beetles are able to discriminate between different types of prey and apply different behavioural tactics to hunt it. As the particular strategies are used to increase hunting success, and as a result allow to accumulate energy for future activity of the predator,

Corresponding author
Radomir Jaskuła,
radomir.jaskula@biol.uni.lodz.pl

it can be expected that such a type of hunting behaviour is characteristic also of other tiger beetle species.

## INTRODUCTION

Tiger beetles (Coleoptera: Cicindelidae) are small to medium-sized predatory beetles hunting for a variety of small, mostly typically epigeic invertebrates. Most species of these fast running predators are characterised by diurnal activity (*Pearson & Vogler, 2001*). Although they typically use sight to locate their fast moving prey (*Świecimski, 1956*; *Gilbert, 1987*; *Gilbert, 1997*), sometimes even day active species can capture prey in complete darkness, which suggests that other senses, such as chemoreception, audioreception and mechanoreception, may play an important role in searching for prey in this beetle group (*Riggins & Hoback, 2005*). A large spectrum of prey, including e.g., Coleoptera, Hymenoptera, Orthoptera, larvae of Lepidoptera, but also spiders or small crustaceans, indicates that these beetles are opportunistic hunters (*Larochelle, 1974*; *Pearson, 1988*; *Pearson & Vogler, 2001*). They may also consume plant material as food (*Hori, 1982*; *Hill & Knisley, 1992*; *Jaskuła, 2013*). Although the diet of tiger beetles as a group is rather well known, little is known about prey preferences and/or hunting strategies of most species. Generally, a tiger beetle locates its live prey visually and after that starts to pursue it in the course of active running interspersed with pause-and-look behaviour (*Gilbert, 1987*; *Gilbert, 1997*) or the beetle waits in a shaded area and attacks the prey when it is approaching (*Kaulbars & Freitag, 1993*). *Pearson & Knisley (1985)* have observed that if the attack is successful, the beetle grabs the prey with its mandibles. Before the prey is consumed, the beetle starts to test it in terms of size, hardness, and noxious chemicals. When the prey is too large and/or is inedible because of some chemical substances, it is quickly released. Moreover, *Świecimski (1956)* has noted that tiger beetles use memory of the shape and location of prey to distinguish small prey located at a shorter distance from large prey placed at a greater distance.

Flexibility in terms of hunting strategies usually brings a significant benefit to the predator. Predatory species which use different behavioural tactics can feed on a larger variety of food, and as a consequence, they can survive in habitats with a low number of specific prey (= higher adaptation rates to environmental change) and/or colonise new areas (= higher dispersal ability). Moreover, individuals of such opportunistic species can more easily and more rapidly accumulate energy needed during the reproduction process, which is especially important for females (*Curio, 1976*).

The aims of this study were: (1) to verify if tiger beetles prefer particular type and size of prey, with different ability to escape, (2) to determinate if tiger beetles use different hunting strategies when attacking prey of different sizes and abilities to escape, and (3) to verify if sex of the studied species makes a difference in its hunting behaviour. Since in

most Cicindelidae females are larger, it is expected that females would prefer to hunt for larger prey than males.

## MATERIAL AND METHODS

### Predator

To test our hypotheses, we have chosen two tiger beetle species. *Calomera littoralis nemoralis* (Olivier, 1790), one of the most common Cicindelidae species in the Mediterranean region, having one of the widest habitat ranges among all tiger beetles known from this region (*Wiesner, 1992*; *Jaskuła, 2011*; *Jaskuła, 2015*; *Jaskuła & Rewicz, 2015*; *Jaskuła, Rewicz & Kwiatkowski, 2015*; *Jaskuła et al., 2016*). Specimens were collected in the mouth of the Evros River (40°49′9.29″N, 25°59′28.59″E) on the Greek marine sandy beach. *Cicindela hybrida hybrida* Linnaeus, 1758, the most common tiger beetle species known from Central Europe, also recognised as an eurytopic species according to habitat types (*Wiesner, 1992*; *Jaskuła, 2005*). Specimens were collected in Krzywie (51°51′26.49″N, 19°26′48.18″E) in an old gravel pit in Central Poland. Adult beetles from both species were collected with an entomological net in August 2008.

A few dozen males and females from both species were caught and taken to the laboratory. Specimens were kept separately in transparent plastic containers with 2-cm layer of sand at the bottom. The proper humidity was maintained through regular water spraying. The experiment was carried at the stable temperature of 24 °C and in the natural photoperiod. Beetles were fed daily by ants (*Lasius niger*) during the whole week before experiment 1.

### Prey

Based on earlier personal observations in the field, six taxa of common prey of tiger beetles were chosen for the study. These were: larvae of grasshoppers (Orthoptera: Acrididae), *Bembidion lampros/B. properans*, *Calathus melanocephalus*, *C. fuscipes* (Coleoptera: Carabidae), and larvae of Symphyta (Hymenoptera). *Bembidion lampros* and *B. properans* were considered as one type of prey due to their similar weight and size and because of difficulties in correct identification of species when the beetle is alive and fast moving. Caterpillars of Symphyta correspond to the next three stages of their development (Table 1). As different types of prey possess different abilities to escape, we have divided them into three groups: (1) Orthoptera—possess a high escape potential as they have jumping legs and can jump a long distance away; (2) ground beetles (Carabidae)—have a medium escape potential as they can run fast and dodge, or turn over, additionally they have the ability to exude a chemical weapon in emergency situations; (3) caterpillars—are unable to move quickly or dodge and turn over so they are characterised by a small escape potential (Table 1). Caterpillars and grasshopper larvae were collected in the field by entomological net, and carabids by exhauster. Different types of prey were collected on a day before the experiment, and stored individually in a refrigerator in order to reduce their mortality.

### Experimental procedure

We conducted experiments in 20-cm diameter plastic buckets with a 2-cm layer of sand at the bottom. Each individual was kept separately. All experiments were conducted between

**Table 1  Prey taxa used in the experiments.**

| Prey species | Order/Family | Size type | Ability to escape | Length (mm) | Mean | Weight (g) | Mean |
|---|---|---|---|---|---|---|---|
| grasshopper | Orthoptera | – | high | – | – | – | – |
| *Bembidion lampros/properans* | Carabidae | small | medium | 3–5 | 4.0 ± 0.44 | 2–5 | 3.8 ± 0.88 |
| *Calathus melanocephalus* | Carabidae | medium | medium | 6–9 | 7.5 ± 0.68 | 8–28 | 16.5 ± 4.12 |
| *C. fuscipes* | Carabidae | large | medium | 9–11 | 9.9 ± 0.55 | 36–68 | 49.4 ± 7.46 |
| Symphyta | Hymenoptera | small | low | 8–12 | 10.4 ± 1.13 | 17–40 | 28.2 ± 4.27 |
| Symphyta | Hymenoptera | medium | low | 12–15 | 13.8 ± 0.81 | 32–63 | 46.4 ± 8.62 |
| Symphyta | Hymenoptera | large | low | 15–21 | 17.3 ± 1.25 | 55–97 | 77.0 ± 11.9 |

**Table 2  Number of repetitions, and types of the prey in each experiment.**

| Species | Experiment 1 | | Experiment 2 | | | | Experiment 3 | | | |
|---|---|---|---|---|---|---|---|---|---|---|
| | | | Caterpillar | | Beetle | | Caterpillar | | Beetle | |
| | N | Type | N | Size | N | Size | N | Size | N | Size |
| ♀♀ *Calomera littoralis* | 47 | gra, m bet, m cat | 48 | s cat, m cat, l cat | 47 | s bet, m bet, l bet | 38 | s cat | 38 | s bet |
| ♂♂ *Calomera littoralis* | 76 | gra, m bet, m cat | 60 | s cat, m cat, l cat | 44 | s bet, m bet, l bet | 38 | s cat | 38 | s bet |
| ♀♀ *Cicindela hybrida* | 70 | gra, m bet, m cat | 53 | s cat, m cat, l cat | 53 | s bet, m bet, l bet | 32 | l cat | 38 | m bet |
| ♂♂ *Cicindela hybrida* | 55 | gra, m bet, m cat | 58 | s cat, m cat, l cat | 46 | s bet, m bet, l bet | 36 | l cat | 38 | m bet |

Notes.

abbreviations: *N*, number of repetition; gra, grasshopper; s bet, small beetle; m bet, medium beetle; l bet, large beetle; s cat, small caterpillar; m cat, medium caterpillar; l cat, large caterpillar.

10:00 and 14:00 h under natural light, during the highest hunting activity of tiger beetles. Experiment 1 started after 1 day of hunger. The next experiments were carried out in a specific order: the finish of Experiment 1, feeding in next day, one day of hunger, day with Experiment 2, feeding in next day, one day of hunger and Experiment 3.

In each experiment, each specimen was used only once.

## Types of experiment
### Prey escape potential
Each tiger beetle (both species, both sexes) got three different types of prey dropped into the experimental bucket at the same time. Preferences of the prey type were recorded after the prey was caught and eaten. Feeding happened in the mornings (10:00) or at noon (12:00), and after two hours the buckets were examined to check which prey was caught and eaten. The types of prey represent different escape potentials: low—medium caterpillar of Symphyta, medium—*Calathus melanocephalus*, high—grasshopper. The number of repetitions differs between species and/or sexes (Table 2) because a different number of species and sexes were collected, and if a tiger beetle specimen refused to attack the prey three times (three periods of two hours' feeding), it was eliminated from this experiment.

### Prey size
Experiment 2 consisted of two parts (a and b). Each specimen (both species, both sexes) got three carabid beetles (part a) or three caterpillars (part b) of different size (Table 2)

dropped into the experimental bucket at the same time. In both cases preferences of the prey size were noticed after the prey was caught and eaten, while feeding and checking were conducted under same conditions as in Experiment 1. In particular, random number of prey individuals measured are presented in Table 1.

### Hunting strategies

In Experiment 3, we checked if tiger beetles use different hunting strategies for different prey types. Specimens (both species, both sexes) were observed separately and every step of their hunting behaviour was noted on special work cards. On the cards, we included each major step of hunting sequences. Types and sizes of prey that were used in this step of the experiment were a result of the two previous experiments and included the most preferred choices of each species and sex of tiger beetle (Table 2).

## Data analysis

In Experiment 1 and 2, we counted the number of repetitions and drew diagrams of preferences in terms of type and size of prey. We interpreted number of repetitions as attack for one from three provided preys in Experiment 1 and Experiment 2 (one of three types of prey in Experiment 1 and Experiment 2, respectively). Moreover, in Experiment 2 correlation between predator body size and prey body size was measured. In Experiment 3, each step of the hunting strategy was checked for significant differences between the species and sexes. The frequency of particular steps was calculated as a percentage of such behaviour in relation to all possible behaviours between two successive stages of hunting, and was indicated by the width of the line in the diagram and by the number above the line. The sequence of the hunting strategy was analysed and the frequency of key steps in each species and strategy was calculated.

Preferences regarding the type and size of prey for both sexes and species of tiger beetles as well as key steps of the hunting sequence were analysed using Pearson's chi-squared test.

## RESULTS

### Prey escape potential

*Calomera littoralis* males and females chose most often caterpillars (91% to 98% respectively), then ground beetles (8% to 2%), and almost ignored grasshoppers (1% to 0%) (Fig. 1). *Cicindela hybrida* males and females chose caterpillars (49% to 44%), almost the same often as ground beetles (51% to 52%) and least willingly caterpillars (0% to 4%) (Fig. 1). There is no dominant type of prey for males and females of *C. hybrida* ($\chi 2 = 0.260$, $df = 1$, $p = 0.05$). However, both sexes completely ignored grasshoppers as the prey and selecting beetles and caterpillars with practically equal frequency. Preferences of the prey type between *C. hybrida* and *Calomera littoralis* differ significantly ($\chi^2 = 65.18$, $df = 1$, $p = 0.05$).
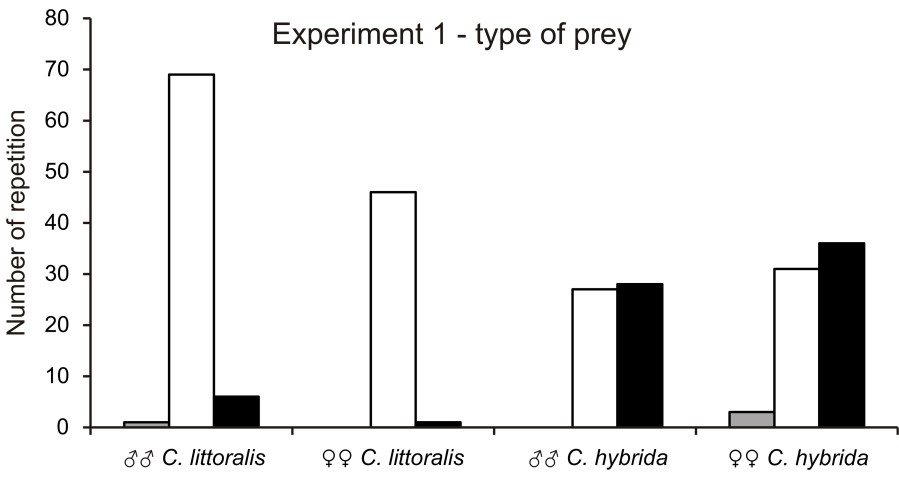

**Figure 1** **Number of chosen preys by males and females of *Calomera littoralis* and *Cicindela hybrida* respectively in experiment 1.** Colors of vertical bars are showing respectively: grasshopper - grey, caterpillar - white, beetle - black.

## Prey size
### Size preferences—carabid beetles
*Calomera littoralis* males and females chose a small beetle (91% to 100%), and less often medium beetle (9% to 0%). *Cicindela hybrida* males and females chose small beetles (30% to 57%), then medium beetle (66% to 43%) and least willingly large beetles (4% to 0%) (Fig. 2A). The size of the preferred beetle prey was significant between the sexes of *C. hybrida* ($\chi^2 = 6.830$, $df = 1$, $p = 0.05$). Preferences of the beetle prey size between *C. hybrida* and *Calomera littoralis* differ significantly ($\chi^2 = 54.522$, $df = 1$, $p = 0.05$).

### Size preferences—caterpillars
*Calomera littoralis* males and females chose small caterpillars (51% to 52%), medium caterpillars (27% to 25%) and large ones (22% to 23%) (Fig. 2B). There were no significant differences between the sexes of *C. littoralis* and the preferred caterpillar size ($\chi^2 = 0.047$, $df = 2$, $p = 0.05$). *Cicindela hybrida* males and females chose small caterpillars (22% to 19%), medium ones (31% to 34%), and large ones (47% to 47%). There were no significant differences between the sexes of *C. hybrida* and the preferred caterpillar size ($\chi^2 = 0.243$, $df = 2$, $p = 0.05$). Preferences of the caterpillar prey size between *C. hybrida* and *Calomera littoralis* differ significantly ($\chi^2 = 25.062$, $df = 1$, $p = 0.05$).

### Predator–prey body-size ratios
Males of both studied tiger beetles species were characterized by constant value of prey length in relation to length of predator where, both beetles and caterpillars were used as prey. In contrast, in case of females constant value of prey body-size was noted only in case of *C. littoralis* hunting on beetles. In females of the some species hunting on caterpillars positive correlation between predator and prey body-size was found, while in females of *Cicindela hybrida* negative correlation was noted for both types of studied preys (Fig. 3).

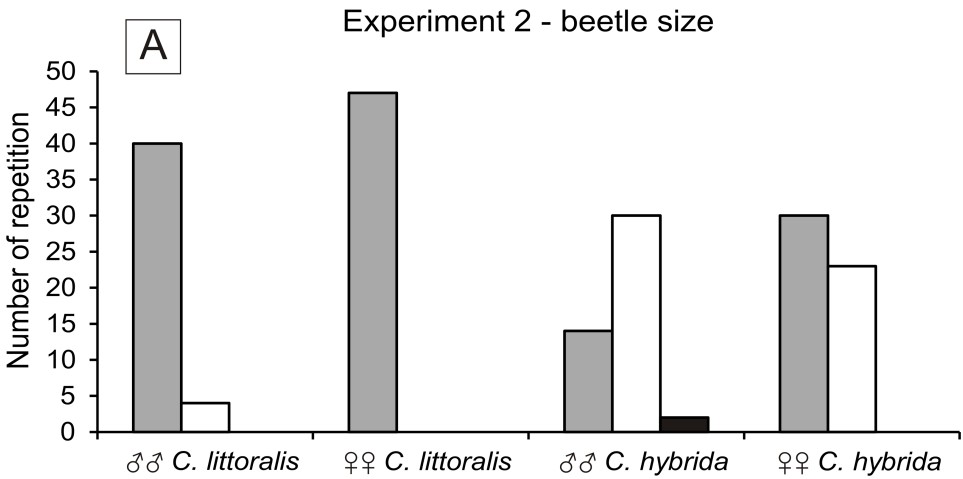

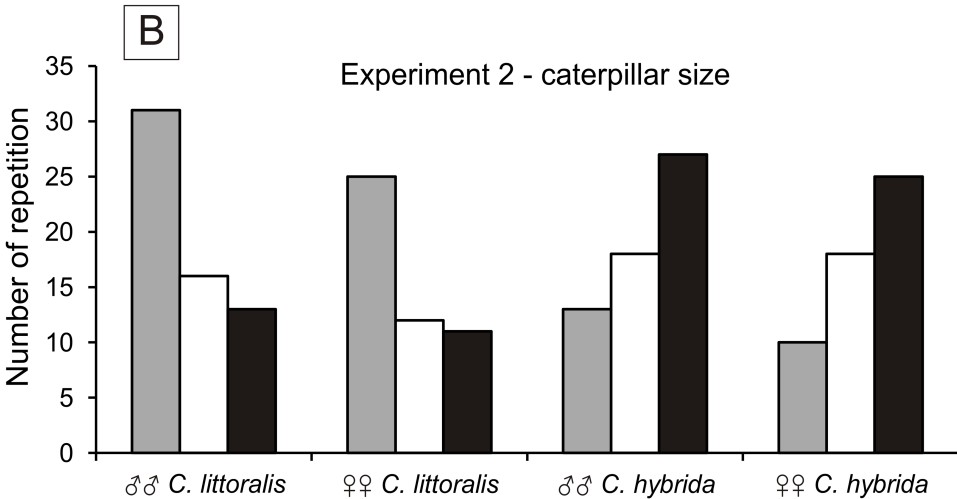

**Figure 2** **Number of chosen preys by males and females of *Calomera littoralis* and *Cicindela hybrida* in Experiment 2 for beetles and caterpillars in (A) and (B), respectively.** In both cases colors correspond to size of the prey in the following pattern: grey, small; white, medium; black, large.

### Hunting strategies

We tested if there were differences between sexes of each species in each major step of the hunting scenario. In most cases, we found there were no differences between the sexes, and we decided to simplify the results of Experiment 3 and to summarise repetitions of both sexes of each species.

### Scenario of hunting prey with different escape potentials

Regardless of the type of prey, the first steps of the hunting pattern were the perception of the prey, followed by the turning of the hunter toward the prey. Next the tiger beetle freezes for a moment (stops), and starts to chase the prey fast in the case of beetles, or nobble the prey slowly in the case of slow caterpillars. The mandible attacks were conducted against
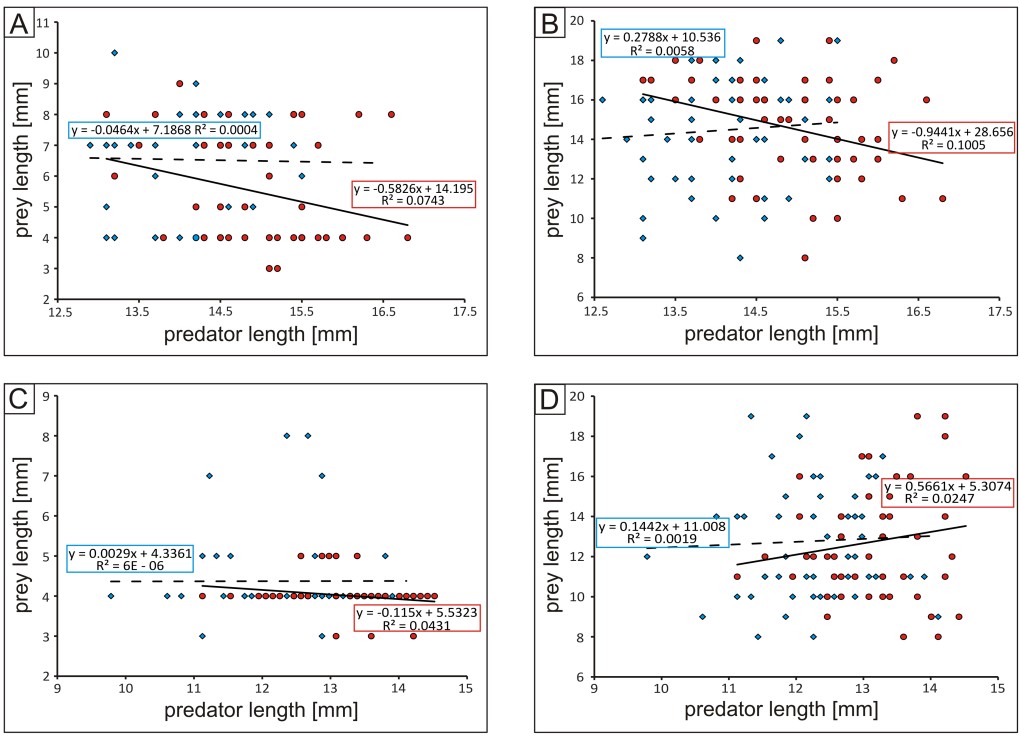

**Figure 3** **Plot of predator-prey body-size ratios: (A)** *Cicindela hybrida* **vs. beetles, (B)** *C. hybrida* **vs. caterpillars, (C)** *Calomera littoralis* **vs. beetles, (D)** *C. littoralis* **vs. caterpillars.** In all cases blue diamonds and dashed line correspond to males and red circles and solid line to females.

three parts of the prey body: front, middle and back. In the case of beetle and caterpillar prey that meant: the front part—head, or the initial sections of the thorax; the middle part the narrow part between the pronotum and the abdomen—the final sections of the thorax or the initial sections of the abdomen; the back part—the abdomen (respectively). We can observe that both species preferred to attack caterpillars in the front or middle part of the body, and avoid the back part. Tiger beetles preferred attacking in its middle part or—less often—the back part, when hunting for fast moving prey. We noticed only one attack on the front part of a beetle (Fig. 4C). After the attack (stabbing with the mandibles), the hunters followed two scenarios; either the attack was lethal and immediately after they ate the prey, or the prey managed to escape after the first stab. After releasing the prey, the hunters mostly retried the attack (re-attack), even repeatedly to achieve the lethal effect. Less often the hunters abandoned (abandonment of the prey) the dead prey, or finished the attack by leaving the wounded prey (ineffective attack). Sometimes after eating the prey only partially, or after an ineffective attack, they would abandon the prey and start digging the ground with mandibles.

### Behavioural prey-type specificity

The beginning of the hunting strategy of *Cicindela hybrida* towards beetles and caterpillars looks similar—after visual prey perception, the hunter turns toward the prey and after

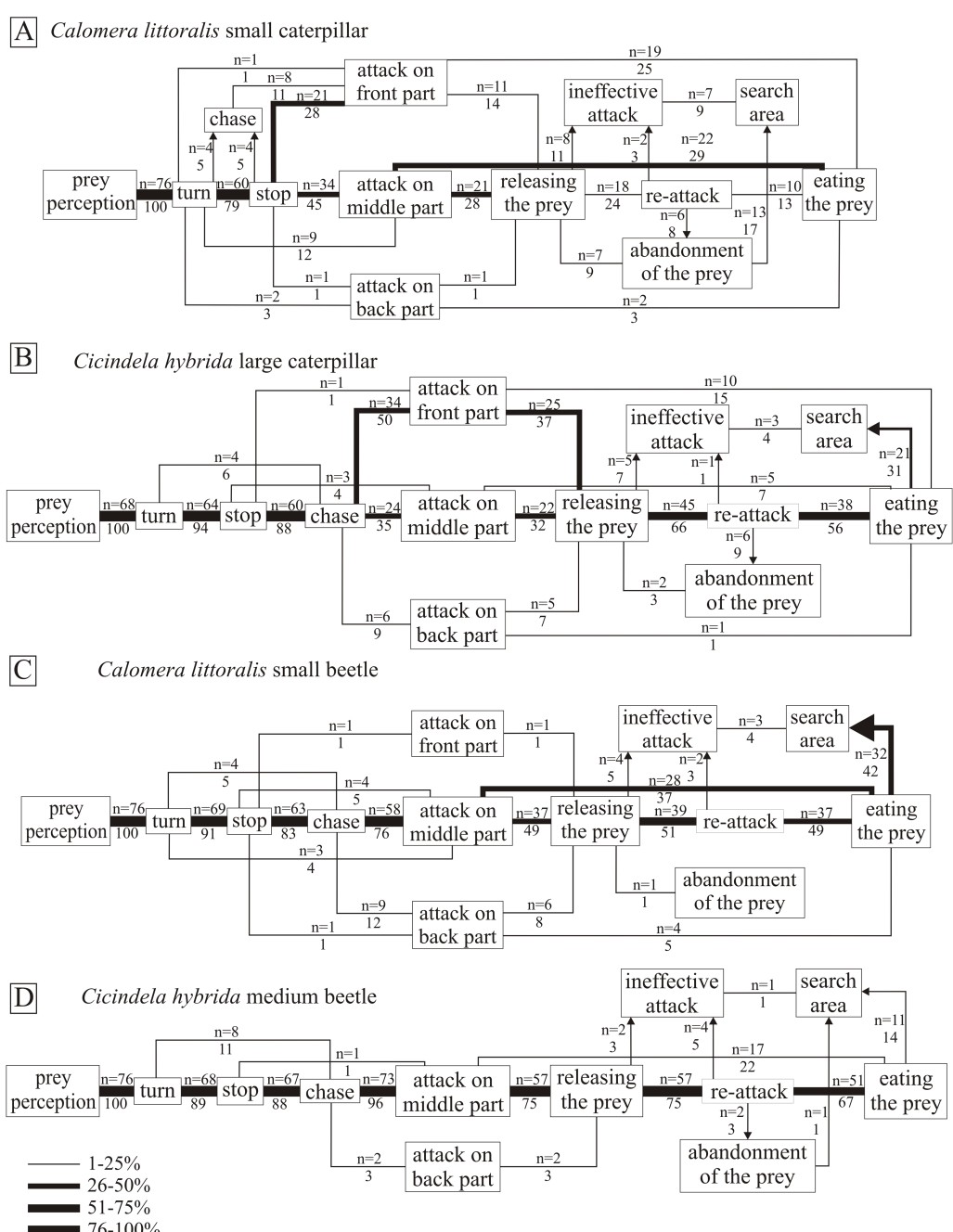

**Figure 4 The flow diagrams of *Calomera littoralis* hunting small caterpillar (A), small beetle (C); and *Cicindela hybrida* hunting large caterpillar (B), and medium beetle (D).** The frequency of particular steps of the hunting strategy was calculated as a percentage of such behaviour in relation to all possible behaviours between two successive stages of hunting, and was indicated by the appropriate line width in the diagram and by the number below the repetition number of such behaviour. The sequence should be read from left to right unless indicated by an arrow.

a moment of observation chases and stabs the prey. The main difference between the strategies concerns the site of the attack. Almost all attacks towards beetles (96%) were conducted in the middle part of their body, in the case of caterpillars the back part was less preferable (9%), most favourable were the front part (50%) and the middle part (35%) ($\chi^2 = 55.18$, $df = 2$, $p < .001$) (Figs. 4B and 4D). After the first attack, the prey was released and attacked again. Caterpillars were abandoned more often than beetles (12% to 3% respectively) ($\chi^2 = 4.63$, $df = 1$, $p < .05$). The last stage of the hunting strategy was also different, *C. hybrida* searched the area more often after hunting caterpillars (37% to 17%) ($\chi^2 = 7.14$, $df = 1$, $p < .05$).

The hunting strategy of *C. littoralis* against beetles and caterpillars shows more differences than similarities (Figs. 4A and 4C). The first clear difference is a lack of chase stage in the case of caterpillars (11% to 88% when attacking beetles) ($\chi^2 = 91.62$, $df = 1$, $p < .05$). The attack against beetles was conducted in 76% in the middle part of the body, and against caterpillars in the front part (39%) and the middle part (57%), which was a significant difference between those types of prey ($\chi^2 = 38.95$, $df = 2$, $p < .00001$). After the first attack, we can observe quite a high level of killed and eaten prey (42% for beetles and 57% for caterpillars), but still there was no significant difference between types of prey. However, killed caterpillars were abandoned more often than beetles (17% to 3% respectively) ($\chi^2 = 8.95$, $df = 1$, $p < .05$). Altogether it indicates the predator's greater efficiency when hunting beetles (91% killed and eaten prey, compared to 70% killed and eaten caterpillars). *Calomera littoralis* searches the area more often after hunting beetles than caterpillars (46% to 26%) ($\chi^2 = 6.41$, $df = 1$, $p < .05$).

### *Behavioural hunter-species specificity*

The hunting pattern for beetles was quite simple for both hunter species. The main attack sequence was straightforward: prey perception, turn, stop, chase, attack on the middle part of the body, releasing the prey, re-attack, eating the prey. Deviations from this pattern were not abundant. We can observe that tiger beetles clearly prefer attacking the middle part of the prey (*Calomera littoralis* 76%, *Cicindela hybrida* 96%) (Figs. 4C & 4C), and almost ignore the front and back parts. The difference between tiger beetle species appears after the attack, *Calomera littoralis* in 58% of cases released the prey after the first stab ($n = 44$), and *C. hybrida* in 78% of cases ($n = 59$) ($\chi^2 = 6.78$, $df = 1$, $p < .05$). As a consequence, also the re-attack occurred more often in *Cicindela hybrida* than in *Calomera littoralis* (75% $n = 57$ to 51% $n = 39$ respectively). *C. littoralis* kills faster than *Cicindela hybrida*, we can observe 42% of killed beetles after the first attack, and only 22% in the case of *C. hybrida* ($\chi^2 = 6.78$, $df = 1$, $p < .01$). However, effectiveness of hunting beetles between *Calomera littoralis* and *Cicindela hybrida* was almost identical with 91% $n = 69$ and 89% $n = 89$ of respectively killed and eaten prey. One more curious behaviour occurred much more often in the *Calomera littoralis* pattern. This species searched the area after hunting in 42% of cases, which is significantly different than 14% of such behaviour instances in *Cicindela hybrida* ($\chi^2 = 14.74$, $df = 1$, $p < .001$).

The hunting strategies towards caterpillars were more complicated than towards beetles. We can observe the first difference between hunter species in chasing or approaching the

prey after turning towards it. *Cicindela hybrida* uses the same pattern as towards beetles - it freezes for a moment and then chases the prey (94%, $n = 64$) (Fig. 4B). Surprisingly, *Calomera littoralis* after freezing approaches slowly (11%, $n = 8$, fast chase) the prey before stabbing ($\chi^2 = 100.31$, $df = 1$, $p < .001$). Both hunters stab the caterpillar mostly in the head or the middle part of the body, less than 10% of attacks were carried to the back part. After the first attack, *C. littoralis* released the prey in 43% of cases ($n = 33$), and *Cicindela hybrida* significantly more often (76%, $n = 52$) ($\chi^2 = 16.21$, $df = 1$, $p < .001$). In consequence, the re-attack occurred only in 24% of cases for *Calomera littoralis* and in 66% of cases for *Cicindela hybrida* ($\chi^2 = 26.33$, $df = 1$, $p < .05$). *Calomera littoralis* has a higher level of success of the first attack and kill, as it happened in 57% of cases and only in 24% of cases the first stabbing by *Cicindela hybrida* resulted in a killed and eaten caterpillar ($\chi^2 = 16.21$, $df = 1$, $p < .001$). However, overall hunting effectiveness was similar, and both hunting species killed and ate more than 70% of their prey.

## DISCUSSION

Although both tiger beetle species used in the experiments can be characterized by similar body size and are known as predators hunting different small arthropods (mainly epigeic insects), occasionally eating also dead insects (*Cicindela hybrida* –*Świecimski, 1956*) or even plant material (*Calomera litoralis* –*Jaskuła, 2013*), the beetles differ in case of other ecological aspects, including geographical distribution and habitat preferences (*Dreisig, 1981*; *Wiesner, 1992*; *Jaskuła, 2011*). That makes both species opportunistic predators hunting for the type of prey which is actually available in the beetle's habitat as it was shown in some other tiger beetle species occurring in different parts of the world (e.g., *Sinu, Nasser & Rajan, 2006*). Moreover, it also may predict significant differences in their hunting strategies and prey preferences (*Brose et al., 2006*). Our results clearly confirm the ability of *Cicindela hybrida* and *Calomera littoralis* to catch and kill different types of prey in terms of body size and mobility. On the other hand, we have noted that in the case of prey mobility, a large number of *C. littoralis* (91 or 98% depending on the beetle sex) and almost half of the studied specimens of *Cicindela hybrida* (44 or 49% depending on the beetle sex) preferred caterpillars which cannot escape faster than fast running beetles. Such a strategy can be clearly explained when the energetic cost of such a predatory behaviour is analysed (*Brose et al., 2008*). From the predator's point of view, predation is a very energy-consuming activity as prey needs to be located, which often takes time, caught and killed, which requires additional energy for a potential fight with the prey, and is often dangerous also for the predator as it can be injured. Also, if the attack is not successful, the predator needs to look for another prey and repeat all the parts of such a behaviour again and again (*Bonsall & Hassell, 2007*; *Creel & Christianson, 2008*). Taking this into consideration, hunting for slow moving prey characterised by a small escape potential is much better as it allows the predator to preserve more energy for any future activity. On the other hand, in the case of *C. hybrida*, fast moving beetles were also noted as very important prey (51 or 52% depending on the beetle sex). This confirms earlier observations by *Świecimski (1956)*, who noted that this species chooses fast moving prey as

their faster movement can be probably easier perceived by the predator. On the other hand, ignoring this type of prey by *Calomera littoralis* (if slow moving caterpillars were available as food) can be probably explained by chemical defence regularly used by different ground beetle species, including *Calathus* and *Bembidion* beetles (*Moore, 1979*), as it is known that tiger beetles often release prey which emits chemicals (*Pearson & Knisley, 1985*). In the case of habitats where *Cicindela hybrida* occurs, both *Calathus melanocephalus* and *Bembidion lampros/properans* are regularly observed, and the tiger beetle was observed hunting for them (R Jaskuła, pers. obs., 2002–2006), while in the case of habitats occupied by *Calomera littoralis*, these species of ground beetles are rarely encountered or even do not occur at all. As a consequence, we cannot exclude the assumption that a lack of potential contact between the prey and the predator under natural conditions does not play a role in choosing the prey under laboratory conditions. As experiments were made in small containers with a flat surface of substrate at the bottom, we can exclude the role of target elevation in prey selection by tiger beetles as was suggested by *Layne, Chen & Gilbert (2006)*. Also, the role of the temperature, a factor noted as important in tiger beetles hunting in the wild (*Dreisig, 1981*), can be ignored as all the experiments were made under the same conditions.

Although tiger beetles can modify their diet according to types of prey which are actually available in the habitat (*Sinu, Nasser & Rajan, 2006*), the size of prey is the second important parameter playing a crucial role in hunting success of the predator (*Alcock, 1993*; *Brose et al., 2006*; *Brose et al., 2008*; *Ball et al., 2015*; *Kalinkat et al., 2011*). Generally it is known that at least in case of some predators, including different beetles (e.g., *Vucic-Pestic et al., 2010*; *Kalinkat et al., 2011*), the majority of predators actively prefer large prey, even if predator–prey body-size ratios vary across different habitats and predator and prey types and mean consumer–resource interaction strengths may be correlated with different habitat parameters and consumer types (*Brose et al., 2006*). Comparing with literature data we believe that also our results suggest that predator–prey body-size ratios can be important in case of both studied tiger beetle species (Fig. 3). We have noted that *Calomera littoralis* preferred small prey with a small (caterpillars) and fast (ground beetles) escape potential (51–100% depending on the beetle's sex and type of prey), while the medium-sized prey was chosen only in the case of slow moving caterpillars. A different situation was observed in the case of *Cicindela hybrida*. In this species, medium (43 or 66% depending on the beetle sex) and small-sized prey (43% in the case of females) was chosen only in the case of fast running ground beetles, while in the case of slow caterpillars much bigger individuals were attacked (47% for both beetle sexes). The body length of both studied tiger beetle species is similar. On the other hand, *C. hybrida* has longer mandibles (up to 10% in females compared with *Calomera littoralis*; *Jaskuła, 2005*), the elements of mouthparts which play a key-role in catching and cutting the prey. Such a difference in the length of mandibles can explain the preference for bigger prey by *Cicindela hybrida*, especially in females, as it is known that longer mandibles allow them to keep a wider distance between the end parts of these organs when the mandibles are fully opened, and as a result potentially bigger prey can be caught (*Pearson & Mury, 1979*). As mentioned above, hunting for bigger and easy to catch prey has great evolutionary sense from the predator's point of view, as such a strategy allows to keep energy for future activity of the predator (e.g., *Brose et al., 2006*; *Brose et al.,*

*2008*; *Ball et al., 2015*; *Kalinkat et al., 2011*). This seems to be especially important in the case of females which need to accumulate much more energy for the breeding season than males (e.g., for production of eggs and finding the right place to lay them) (*Thornhill & Alckock, 1983*).

Both studied tiger beetle species located their prey visually and then tried to catch it after fast active pursuit interspersed by short stops. All these elements of hunting behaviour were earlier noted in *C. hybrida* (*Świecimski, 1956*) as well as in other tiger beetle species (e.g., *Gilbert, 1986*; *Gilbert, 1987*; *Gilbert, 1997*) and seem to be very typical for all beetles classified in this group, even if at least some diurnal species can locate and catch prey without sight (*Riggins & Hoback, 2005*). Although there are no data about the physiological base of such a pause-and-look behaviour in the case of the species studied by us, it is known that in other tiger beetles such a behaviour plays a very important role in the actualisation of prey position as the signal sent from ommatidia in the beetle's eyes to its central nervous system is slower that the speed of running tiger beetle (*Gilbert, 1997*). As in the cases of earlier studied tiger beetle species (e.g., *Świecimski, 1956*; *Pearson & Knisley, 1985*; *Gilbert, 1987*; *Lovari et al., 1992*; *Gilbert, 1997*; *Zurek, Perkins & Gilbert, 2014*), we have noted that *Calomera littoralis* and *Cicindela hybrida* use mandibles to test the size, shape, and probably also noxious chemicals of their prey before it is killed and eaten. The significant difference in the "testing behaviour" observed by us between both species (releasing of prey in 58% cases in *Calomera littoralis* and 78% in *Cicindela hybrida*) is most probably connected with the size of their preferred prey. Smaller prey can be faster and more easily tested that the bigger one, and as a consequence can be killed quicker. Exactly such an observation was noted in the case of *Calomera littoralis* which preferred smaller types of prey. On the other hand, *Cicindela hybrida*, which hunted mainly medium and/or larger prey, was characterised by a much longer "testing behaviour" (prey released in 75–76% of cases after the first attack). Such a behaviour seems to play an important role as the final hunting success was very similar in both species.

Both tiger beetles clearly preferred attacking in the middle (the connection between the thorax and the abdomen in ground beetles) or in the middle or front part of the prey (the head or the thorax in caterpillars), almost ignoring the back parts. The explanation of such a hunting strategy is rather simple as the main muscles responsible for walking are places in the insect's thorax. Moreover, in the front part of the insect body (the head and partly the thorax), the main part of the insect nervous system in placed (*Gilliot, 2005*). As a consequence, an attack on these body parts usually allows to immobilise and kill the prey quickly. Although there is only a small number of studies on the hunting behaviour of tiger beetles, and therefore we cannot provide similar results from the literature, single field observations of the second author upon some European (*Cephalota chiloleuca*, *C. circumdata*, *Cicindela campestris*, *C. sylvatica*, *Cylindera germanica*, *C. trisignata*, *Myriochila melancholica*) and North African species (*Grammognatha euphratica*, *Lophyra flexuosa*) suggest that this is a common strategy among tiger beetles. Moreover, the same or very similar strategy can be found in other predatory insects which need to catch prey quickly, such as some diurnal ground beetles (e.g., *Bauer, 1981*; *Bauer, 1985*) as well as other

terrestrial arthropods, including jumping spiders (e.g., *Jackson & Pollard, 1996*; *Bartos, 2002*; *Bartos, 2007*; *Bartos, 2008*; *Bartos & Minias, 2016*).

## CONCLUSIONS

The results of the presented study clearly confirm that the hunting behaviour of tiger beetles is complicated and multi-staged. *Calomera littoralis* and *Cicindela hybrida* are able to discriminate between different types of prey (both according to their size and escape potential) and apply different behavioural tactics to hunt them, what most probably is connected with predator–prey body-ratio. Particular strategies are used to increase hunting success and as a result allow to accumulate energy for future activity of the predator. Although there is a lack of similar data for most of other tiger beetle genera, we should expect that this type of behaviour, very logical in the wide evolutionary sense, is characteristic for the entire group. On the other hand, future studies, especially on nocturnal and/or arboreal tiger beetle species which occupy different types of environment or hunt at night, may provide additional facts about hunting strategies of Cicindelidae.

### Funding
This work was financed partially from the internal funds of the University of Lodz. There was no additional external funding received for this study. The funders had no role in study design, data collection and analysis, decision to publish, or preparation of the manuscript.

### Grant Disclosures
The following grant information was disclosed by the authors:
University of Lodz.

### Competing Interests
The authors declare there are no competing interests.

### Author Contributions
- Tomasz Rewicz conceived and designed the experiments, performed the experiments, analyzed the data, contributed reagents/materials/analysis tools, prepared figures and/or tables, authored or reviewed drafts of the paper, approved the final draft.
- Radomir Jaskuła conceived and designed the experiments, analyzed the data, contributed reagents/materials/analysis tools, authored or reviewed drafts of the paper, approved the final draft.

### Data Availability
The raw data are provided in the Supplemental Files.

## Supplemental Information

Supplemental information for this article can be found online at http://dx.doi.org/10.7717/peerj.5971#supplemental-information.

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
