# Peer review of "Catch fast and kill quickly: do tiger beetles use the same strategies when hunting different types of prey?"

_PeerJ, doi:10.7717/peerj.5971_

## Round 0.1 · original submission · Major Revisions

Three expert reviewers, with specific knowledge of beetle ecology and behaviour, have provided detailed assessments of the paper. Whilst all three referees see much merit in the research, they have offered detailed advice on how the paper can be revised. There is a particular emphasis on improving the language and flow of the paper, and on the need to reduce wordiness. I would therefore like to invite the authors to undertake a major revision for reconsideration, in which all of the referee's concerns are addressed or at least answered appropriately. Please also aim to reduce the length of the paper by at least a third - tightening the prose in this way will make for a much more readable final product.

Reviewer 1 ·

Basic reporting

Basic Reporting. In summary, this manuscript presents useful, valuable and new information about tiger beetle feeding, but is very poorly written and thus requiring extensive revision. The value of the paper is that the study addresses new information about prey selection and feeding processes of tiger beetle feeding, and something which is largely absent in the literature. The introduction and discussion provide valuable and complete review of the literature, indicating the authors know what has and has not been done on the subject. The introduction is generally well organized and complete.
Undoubtedly the poor writing is related to difficulty in writing in English, but still the presentation in much to wordy and must be rewritten in a more concise and direct manner. This is the case for all parts of the paper. In the attached version, I include many track changes which I believe provides guidance on proper presentation. These track changes are included in the Introduction, Methods and Results but not the discussion which is also in need of significant revision. There is also some inappropriate use of terminology, such as using front, middle, back rather than accurate anatomical body regions.
The figures and tables are well done and appropriate for supporting the results of this study. The captions do need to be redone, however. Figure 3 is a novel and excellent representation of the hunting and feeding sequence of behaviors, and I applaud the authors for this creation. It is valuable part of the paper.

Experimental design

I think the experimental design is in general adequate, especially the size selection studies. The experiment dealing with prey escape differences is probably acceptable and difficult to test by any design. The selection and capture/feeding on these three different prey types cannot be concluded solely due to their escape behavior since these prey types are so different in many ways. Thus, the experiment in useful only testing selection of different prey types. A design that immobilizes the faster prey might be an approach, such as removing legs of beetles or grasshoppers compared to normal ones. The simple statistics used in the analysis are adequate.
The discussion includes a good review of possibly explanations of the observed results, bringing in relevant literature but needs to be rewritten to eliminate a repeat of results. ,. The explanations of differences seem to be accounted for. I do think the discussion like the rest of the paper could be rewritten more concisely. And, I think it would benefit from reorganization by first detailing the differences in the two tiger beetles and their habitats and then follow with how these could explain the differences observed.
To conclude, I think this manuscript makes significant positive contributions to tiger beetle biology, but must be rewritten to be acceptable before publication.

Validity of the findings

Nothing to add

Additional comments

As indicated in the review, this manuscript needs a major rewrite to more adequately conform to proper scientific writing.

Annotated reviews are not available for download in order to protect the identity of reviewers who chose to remain anonymous.

·

Basic reporting

Clear, unambiguous, professional English language used throughout.Yes
Intro & background to show context. Yes
Literature well referenced & relevant. Yes
Structure conforms to PeerJ standards,
discipline norm, or improved for clarity. Yes
Figures are relevant, high quality, well labelled & described. Yes
Raw data supplied. Yes

Experimental design

Original primary research within Scope of the journal. Yes
Research question well defined, relevant & meaningful. Yes
It is stated how the research fills an identified knowledge gap. Yes
Rigorous investigation performed to a high
technical & ethical standard. Yes
Methods described with sufficient detail & information to replicate. Yes

Validity of the findings

Data is robust, statistically sound, & controlled. Yes
Conclusions are well stated, linked to original
research question & limited to supporting results. Yes

Additional comments

I think the sentences in lines 147-148 of the subsection “Prey size”, in lines 185-186 of the subsection “Prey escape potential”, in lines 198-199 of the subsection “Size preference – carabid beetles”, in lines 207 and 213 of the subsection “Size preference – caterpillars” can be deleted, because this information was given earlier.

Text in lines 151-154 can be modified as follows:
“There were 56 random individuals in each size of caterpillars measured and weighted (Table 1).
In both cases preferences of the prey size were noticed after the prey was caught and eaten, while feeding and checking were conducted under same conditions as in Experiment 1”.

The names of carabid beetles in lines 195-197 of the subsection “Size preference – carabid beetles” are unnecessary.

In subsection “Prey escape potential” is indicated that in the case with Calomera littoralis the caterpillars were the most common type of prey for both sexes, while in the case with Cicindela hybrida the dominant type of prey was not detected, because males and females caught caterpillars and carabid beetles with practically equal frequency. On the one hand both these propositions are true. But on the other hand both species are ignored grasshoppers as the prey. Thus, even in the case with Cicindela hybrida we can say about preference of the prey – caterpillars and carabid beetles in the equal probability.

Please, see these and other corrections in the attached PDF as well.

Reviewer 3 ·

Basic reporting

In this manuscript the authors describe the foraging behavior of two common European tiger beetles towards different types of prey with a particular focus on the predators' preferences in terms of the preys' size and their flight abilities. I found a couple of critical issues that need clarification and I will also give some suggestions how this manuscript could potentially be improved. Below I will post more specific recommendations which I hope the authors will find useful for a revision of their manuscript.
Generally, I feel the level of English language use in this manuscript is sufficient. It could be improved at some points; some suggestions are given below.
One general aspect I want to highlight: in light of recent publications on size-related prey preferences in carabids (Brose et al. 2008, Vucic-Pestic et al. 2010, Kalinkat et al. 2011, Ball et al. 2015) it seems that it would make sense to add the respective information on predator-prey size ratios (or mass ratios) to this manuscript (also see Brose et al. 2006). Thereby it could be better connected to the ecological literature on general patterns and mechanisms that explain many aspects of predator-prey ecology with allometric scaling relationships (also see further comments below).
Another general remark about the whole storyline in the manuscript: for my taste the hypotheses stated in lines 83 to 95 could be much more specific and also better supported through the introduction. I feel at the moment these hypotheses are rather weak and almost trivial. Moreover, connecting the modified (and, ideally three,) hypotheses more specifically to the three experiments would make for a much more coherent manuscript.
Experimental design

Experimental design

I found it quite intriguing and also critical that there was absolutely no information about standardization of hunger levels. It has recently been shown that predator starvation levels in experiments that examine density-dependent feeding are absolutely critical for reliable results (Li et al. 2018) and, more generally, my experience also tells me that this kind of information should be stated and, actually is, usually given in almost all kinds of lab-based predator feeding studies.
Another thing that needs clarification: were all observations of all three experiments performed over the whole experimental time of four hours? This was not entirely clear to me.

Validity of the findings

No comments.

Additional comments

specific comments for the authors:
l.59: this sounds a bit odd, I would say either „chemoreception, audioreception and mechanoreception“ or „smell, hearing or touch“
ls.79/80: rather „... specific prey (= higher adaptation rates to environmental change) and/or colonise new areas (= higher dispersal ability).“
ls. 107-109: this seems like a statement/information that clearly belongs in the funding acknowledgement section at the end of the manuscript.
l.163: what exactly is meant by „the number of repetitions“ here? It is not clear to me how to interpret this. Is it meant to refer to repeated attacks of predators on prey specimen? Then I would recommend stating this explicitly.
ls. 192 ff: as stated in my general comments above I would really appreciate if body sizes, body size ratios and body size preferences could be quantified in this section. It seems for most of the interactions the data to easily provide these informations is there (even if there are only size classes, i.e. average values and no individual based size ratios for each single replicate which would be even better). In the light of recent advancements to understand arthropod food webs and communities based on allometric scaling relationships this would be a much more valuable contribution.
ls. 223 ff: the use of grammatical tense here is inconsistent which impairs comprehension to some extent. Please check throughout text and modify accordingly.
ls. 297 ff: As stated above, including and formalising quantitative information on predator-prey size ratios will help a lot to connect this stduy to the broader ecological literature. This general suggestion applies throughout the discussion and I encourage the authors to reflect on the studies that I suggested above to expand their discussion of predator-prey size relationships of tiger beetles.

References
Ball, S. L., B. A. Woodcock, S. G. Potts, and M. S. Heard. 2015. Size matters: Body size determines functional responses of ground beetle interactions. Basic and Applied Ecology 16:621–628.
Brose, U., R. B. Ehnes, B. C. Rall, O. Vucic-Pestic, E. L. Berlow, and S. Scheu. 2008. Foraging theory predicts predator-prey energy fluxes. Journal of Animal Ecology 77:1072–1078.
Brose, U., T. Jonsson, E. L. Berlow, P. Warren, C. Banasek-Richter, L.-F. Bersier, J. L. Blanchard, T. Brey, S. R. Carpenter, M.-F. C. Blandenier, L. Cushing, H. A. Dawah, T. Dell, F. Edwards, S. Harper-Smith, U. Jacob, M. E. Ledger, N. D. Martinez, J. Memmott, K. Mintenbeck, J. K. Pinnegar, B. C. Rall, T. S. Rayner, D. C. Reuman, L. Ruess, W. Ulrich, R. J. Williams, G. Woodward, and J. E. Cohen. 2006. Consumer-resource body-size relationships in natural food webs. Ecology 87:2411–2417.
Kalinkat, G., B. C. Rall, O. Vucic-Pestic, and U. Brose. 2011. The allometry of prey preferences. PLoS ONE 6 (10):e25937.
Li, Y., B. C. Rall, and G. Kalinkat. 2018. Experimental duration and predator satiation levels systematically affect functional response parameters. Oikos 127:590–598.
Vucic-Pestic, O., B. C. Rall, G. Kalinkat, and U. Brose. 2010. Allometric functional response model: body masses constrain interaction strengths. Journal of Animal Ecology 79:249–256.

---

## Round 0.2 · accepted · Accept

Thank you for taking the time to thoroughly address the critical points of the referees and myself. I think the MS is much improved as a result.

#